# Forgetting the Unforgettable: Transient Global Amnesia Part II: A Clinical Road Map

**DOI:** 10.3390/jcm11143940

**Published:** 2022-07-06

**Authors:** Marco Sparaco, Rosario Pascarella, Carmine Franco Muccio, Marialuisa Zedde

**Affiliations:** 1Division of Neurology with Stroke Unit, Department of Neurosciences, A.O. “San Pio”, P.O. “G. Rummo”, Via Dell’Angelo 1, 82100 Benevento, Italy; marco.sparaco@ao-rummo.it; 2Neuroradiology Unit, Azienda Unità Sanitaria Locale-IRCCS di Reggio Emilia, Via Amendola 2, 42122 Reggio Emilia, Italy; rosario.pascarella@ausl.re.it; 3Neuroradiology Unit, Department of Neurosciences, A.O. “San Pio”, P.O. “G. Rummo”, Via Dell’Angelo 1, 82100 Benevento, Italy; franco.muccio@ao-rummo.it; 4Neurology Unit-Stroke Unit, Azienda Unità Sanitaria Locale-IRCCS di Reggio Emilia, Via Amendola 2, 42122 Reggio Emilia, Italy

**Keywords:** transient global amnesia, amnesia, hippocampus, migraines, memory

## Abstract

Transient global amnesia (TGA) is a clinical syndrome characterized by the sudden onset of a temporary memory disorder with profound anterograde amnesia and a variable impairment of the past memory. Usually, the attacks are preceded by a precipitating event, last up to 24 h and are not associated with other neurological deficits. Diagnosis can be challenging because the identification of TGA requires the exclusion of some acute amnestic syndromes that occur in emergency situations and share structural or functional alterations of memory circuits. Magnetic Resonance Imaging (MRI) studies performed 24–96 h after symptom onset can help to confirm the diagnosis by identifying lesions in the CA1 field of the hippocampal cornu ammonis, but their practical utility in changing the management of patients is a matter of discussion. In this review, we aim to provide a practical approach to early recognition of this condition in daily practice, highlighting both the lights and the shadows of the diagnostic criteria. For this purpose, we summarize current knowledge about the clinical presentation, diagnostic pathways, differential diagnosis, and the expected long-term outcome of TGA.

## 1. Introduction

Transient global amnesia (TGA) is a clinical syndrome characterized by the sudden onset of profound anterograde amnesia and a less prominent retrograde memory impairment, lasting up to 24 h and not otherwise associated with other neurological deficits [1,2,3,4]. Headache, dizziness, and nausea are the most common accompanying complaints [4]. Even if the symptoms of TGA are quite characteristic, the differential diagnosis includes some acute amnestic syndromes, transient and totally reversible, that occur in emergency situations and share structural or functional alteration of memory circuits [5].

In recent years, magnetic resonance imaging (MRI) studies have proven useful in confirming the diagnosis by identifying diffusion-weighted (DWI) lesions in the CA1 field of the hippocampal cornu ammonis [6]. However, the level of detection of the hippocampal DWI lesions in patients with TGA is highly time-dependent, so the maximum diagnostic yield of DWI lesions occurs 24 to 96 h after the onset of symptoms [7,8,9].

The annual incidence of TGA is on average at 3.4–10.4/100,000 and increases to 23.5/100,000 in subjects over 50 years of age [1,3,4,10,11]. The age at onset ranges from 61 to 67.3 years and the gender distribution is estimated at 50.7% females and 49.3% males [12,13,14,15,16].

Although numerous cases of TGA have been described in the literature to date, the diagnosis of this condition still presents pitfalls, especially in the emergency setting where quick decisions have to be made in order to avoid useless or even dangerous treatments. Furthermore, the diagnostic criteria currently in use present lights and shadows that deserve to be highlighted and discussed.

In this review, we aim to provide a smooth and reliable tool for the timely and accurate recognition of TGA in daily practice, summarizing current knowledge about the presentation, diagnosis, and expected long-term outcome.

## 2. TGA Roadmap

Addressing a patient with a suspected TGA requires a precise and detailed roadmap by the neurologist, keeping in mind the diagnostic criteria, the differential diagnosis, the mandatory and optional investigations, and the timing of each of these issues. The main steps of this proposed roadmap are summarized in Figure 1 and detailed in the following paragraphs.

### 2.1. Risk Profile

Over the years, epidemiological studies have shown the association of TGA with some risk factors.

***Migraine history***. In 2014 a large nationwide, population-based cohort study, enrolling 158.301 migraine patients and 158.301 healthy controls (HC), demonstrated that migraines are associated with an increased risk of TGA (incidence rate ratio =2.48, *p* = 0.002), particularly in female patients aged 40–60 years [17]. Noteworthy, in the same study, the subjects with a history of migraines had a significantly younger age of TGA onset (56.6 years) compared to the control group (61.4 years), suggesting that migraines could lead to an earlier age of disease onset [17]. In a recent analysis of the data obtained from the Nationwide Inpatient Sample, which represents 20% of the US community hospitals for the years 1999–2008, patients with a diagnosis of migraines had 5.98 times greater odds of having TGA compared with patients without migraines [18]. In a more recent systematic review and meta-analysis, it was confirmed that there is a higher relative risk (RR) of TGA for migraine vs. non-migraine individuals [RR = 2.48, 95% confidence-interval (95% CI) = (1.32, 4.87)] [19].***Psychiatric comorbidity***. Epidemiological studies suggest that some personality traits might be relevant to the etiology of the disease [7,13,20,21]. Pantoni et al. found that TGA patients had a significantly higher percentage of depression or anxiety disorder, as well as phobic traits in comparison with patients who have had a transient ischemic attack (TIA) or with HC [13]. Additionally, a significantly higher percentage of TGA subjects (33.3%) reported a family history of psychiatric disease as compared with TIA subjects (13.7%) [13]. Other authors have found an increased frequency of psychological or emotional instability and a tendency to feel guilty among patients experiencing TGA events [12,22].***Vascular risk profile***. A retrospective case–control study comparing 293 TGA patients to 632 patients with TIA and 293 age- and sex-matched HC showed a significantly higher prevalence of hyperlipidemia and ischemic heart disease in TGA patients when compared to TIA patients or HC [23]. Conversely, diabetes mellitus was associated with a significantly reduced occurrence of TGA [23]. In a systematic review of observational studies examining the relationship between the conventional cardiovascular risk factors and TGA, there was evidence of a potential association between severe hypertension (defined according to a 160/95 mmHg cut-off) and TGA [24]. Diabetes mellitus (stronger evidence) and current smoking (limited evidence) were found to exert a protective effect [24]. Furthermore, the role of hypertension in TGA was extensively evaluated in a recent analysis that compared the cardiovascular risk profile of 277 patients with TGA to 216 patients with acute ischemic stroke [25]. In this study, patients with TGA had significantly higher systolic and diastolic blood pressure at admission than stroke patients, but lower signs of chronic hypertension, as measured by the extent of cerebral microangiopathy and degree of septal hypertrophy in transthoracic echocardiography [25].

### 2.2. Precipitating Events

Information about precipitating events and the beginning of the attack should be available from a capable observer who witnessed the onset (Table 1).

Documented TGA attacks are preceded in 50–90% of cases by precipitating events that may be divided into the following classes:Emotional stress, (i.e., triggered by medical procedures, interpersonal conflict, birth/death announcement, and difficult/exhausting workday);Physical effort, (i.e., gardening, housework, and sawing wood);Acute pain;Water contact/temperature change, (i.e., hot bath/shower and cold swim);Sexual intercourse;Valsalva-associated maneuvers;Unspecified [6,7,10,12,20,26,27,28].

In a retrospective cohort study of 203 TGA cases from a single center in Buenos Aires, 66% of patients referred to a precipitating event for TGA; most often a Valsalva maneuver (41%) [29].

Quinette et al. found a different profile of precipitating events between the sexes: in men, the TGA episode occurred more frequently after a strenuous physical event, whereas, in women, TGA was often precipitated by an emotional event [12]. The same sex-related differences in the precipitating event profile were found in a recent retrospective observational analysis of 389 patients with TGA [28]. In this study, emotional triggers were experienced more often by women (37.2% vs. 22.8%, *p* = 0.003), while physical stressors were more frequent in men (30.7% vs. 41.1%), *p* = 0.035). In the same study, the emotional trigger was often classified as interpersonal conflict (42.7%) [28]. However, emotional and psychological distress are frequent precipitating events in patients with TGA and are often associated with phobic personality traits [4,12,24,27,30].

### 2.3. Clinical Picture

The clinical presentation of the TGA is characterized by the sudden onset of temporary memory impairment with a prominent inability to form new memories (anterograde amnesia) and a variable impairment of the past memory (retrograde amnesia) [1,2,3,4]. Typically, patients do not fix any novel information, e.g., the treating physician or why they were brought to the hospital, so they repeatedly ask questions, such as, “Why are we here?”, “What time is it?”, or “How did I get here?” [3,31]. Patients remain alert, fully communicative, and keep intact higher cortical functions such as language, calculations, visuospatial skills, reasoning and abstract thinking [32]. Thus, during the episode, they maintain personal and family members’ knowledge, preserve remote memories, and perform previously learned activities, (e.g., driving without impairment) [3,33]. Mild vegetative symptoms, such as headache, nausea, and dizziness may occur during the attack [31]. Chills or hot flushes, fear of dying, cold extremities, paraesthesia, emotionalism, trembling, chest pain, and sweating have also been reported in the literature [4,12].

A recent study on 665 patients determining a chronological pattern of TGA occurrence identified a significant circadian rhythm with a major peak in the morning between 10 and 11 am and a secondary peak between 4 and 5 pm [34].

TGA typically lasts a few hours, often 4 to 6 h, and always less than 24 h [3,12,35]. Short-duration TGA episodes (<1 h) are not uncommon, ranging between 9–32% of cases [1,3,36]. Many studies have shown complete recovery of memory functioning several months after an episode of TGA [6,30,37,38,39,40,41]. However, some authors have reported the persistence of a subclinical impairment of memory functions for months after the acute episode [7,42,43,44,45,46]. Because patients cannot store new memories during the episode, they will have a permanent memory gap for the duration of the attack.

### 2.4. Cognitive Evaluation and Physical Examination during TGA

#### 2.4.1. Main Cognitive Alterations


**
*Reduction in anterograde episodic long-term memory*
**


It consists of a difficulty in learning and subsequently recalling new episodic information after a variable delay [32]. This dysfunction seems to be related to a deficit of both storage and encoding components of episodic memory [47] (see Part I of this review for details).


**
*Partial loss of retrograde episodic long-term memory*
**


Patients have difficulty recalling episodic information learned hours, days, or months before the onset of the amnestic episode. Retrograde amnesia of TGA has been attributed to a difficulty in retrieving information from episodic memory [32].


**
*Reduction of executive function*
**


Executive functions refer to cognitive processes that control and coordinate both cognition and behavior. A meta-analysis of 152 effect sizes from 25 studies showed a “large” reduction in the executive function of TGA patients in relation to comparison subjects [48].

#### 2.4.2. Preserved Cognitive Functions in TGA


**
*Short-term memory*
**


(I.e., reproduction of information without a delay or after a short delay during which the information can actively be tested).


**
*Semantic memory*
**


(I.e., general and acontextual knowledge about the world).


**
*Implicit and procedural memory*
**


(I.e., nondeclarative memory contents such as skills and motor abilities) [32,46].

Since the lack of additional neurological symptoms is mandatory for the diagnosis of TGA (Table 1), a neurological examination should be performed during the attack and not after to be sure that other neurological symptoms and signs do not accompany amnesia.

### 2.5. Diagnostic Criteria

The clinical diagnosis of TGA is confirmed by applying the criteria provided by Hodges and Warlow [2] (Table 1).

Although these criteria still remain valid in clinical practice, some additions are necessary. The possibility of associated retrograde amnesia is not included in the criteria, but it is well recognized that patients with TGA can have some degree of retrograde amnesia during the episode [3,49]. As mentioned earlier, several studies show the involvement of other cognitive functions, such as executive functions or visuo-perceptual abilities [3,44]. Furthermore, the criteria currently adopted, in our opinion, do not allow us to exclude some acute amnestic syndromes that occur in emergency situations, such as ischemic or hypoxic events, migraines, toxic amnesia, etc.

### 2.6. Laboratory Tests and Instrumental Evaluation

No laboratory investigations can actually confirm the diagnosis of TGA. However, a laboratory diagnostic workup should include at least glucose and electrolyte dosage. Hypoglycemia can result in an amnestic deficit and might be considered a differential diagnosis if the patient is diabetic. Alcohol level and a toxicology screen should also be reviewed (see Section 2.8).

#### 2.6.1. Electroencephalography (EEG)

During and after typical TGA episodes, EEG findings have been reported to be normal [11,50,51]. EEG should be considered if there are features suggestive of repetitive or seizure-like etiology. Temporal lobe or complex partial seizures might present in fact as transient epileptic amnesia (TEA), particularly when repetitive episodes of transient amnesia occur [52] (see Section 2.8).

#### 2.6.2. Transthoracic Echocardiography

Echocardiography may be indicated to evaluate left ventricular ejection fraction and septal hypertrophy in TGA patients with elevated blood pressure on admission. In fact, septal hypertrophy, defined as the presence of an increase in the thickness of the septum (women >9 mm, men >10 mm), is considered a possible indicator of chronic hypertension [25]. Therefore, transthoracic echocardiography can help differentiate chronic hypertension from acute hypertension and support the indication of antihypertensive drugs in patients with previously unknown hypertension.

### 2.7. Neuroimaging

Although the diagnosis of TGA is largely clinical and of exclusion, neuroimaging can provide an important diagnostic contribution.

#### 2.7.1. Magnetic Resonance Imaging (MRI)

In 1998, Strupp et al. first described high-signal hippocampal lesions using DWI MRI [53]. Since then, the role of MRI in the diagnosis of TGA has been largely confirmed and clarified [6,8,54,55,56,57]. It has been demonstrated, in fact, that the detection rate of hippocampal lesions in TGA can be improved by up to 85% with optimized MRI parameters and by acknowledging the time course of the lesion [7].

The following are the main MRI findings in patients with TGA (Table 2):Almost all lesions can be selectively found in the area corresponding to the CA1 sector (Sommer sector) of the hippocampal cornu ammonis [6,7,8] (Figure 2).Lesions can be single or multiple and vary in size from 1 to 5 mm [7].In a recent meta-analysis of 1732 patients with TGA the pooled incidence of right, left, and bilateral hippocampal lesions were 37% (95% CI, 29–44%), 42% (95% CI, 39–46%), and 25% (95% CI, 20–30%), respectively [9].In the same study, DWI with a slice thickness ≤3 mm showed a higher diagnostic yield than DWI with a slice thickness >3 mm [63% (95% CI, 53–72%) vs. 26% (95% CI, 16–40%), *p* < 0.01] and there was no significant difference in the diagnostic yield between 3 T and 1.5 T imaging [pooled diagnostic yield, 31% (95% CI, 25–38%) vs. 24% (95% CI, 14–37%), *p* = 0.31)] [9].Neuroimaging data have shown that the level of detection of hippocampal DWI lesions in patients with TGA is highly time-dependent: DWI performed at an interval between 24 and 96 h after symptom onset has a higher diagnostic yield than DWI performed within 24 h or later than 96 h [7,8,9].Focal hippocampal DWI lesions generally resolve 7–10 days after onset of TGA, with no long-term structural changes [58]. This complete reversibility of DWI hippocampal hyperintensity without structural sequelae, as confirmed by the lack of persistent signal change on T2-weighted or FLAIR sequences, does not conform to the time course of classic ischemic lesions [9].T2-weighted and FLAIR sequences allow us to identify and evaluate the extent of cerebral microangiopathy in order to provide a measure of the presence and degree of chronic hypertension [25]. In this way, these sequences can provide useful indications for a more rigorous antihypertensive drug treatment in patients with chronic hypertension and can help to calculate the risk of subsequent TGA recurrence in patients without microangiopathic alterations (for details see paragraph 3.1 of this review).Complementary imaging studies combining MRI and focal MR spectroscopy (MRS) of CA1 DWI/T2 lesions revealed a transient lactate peak without changes of N-acetyl-aspartate (NAA) and creatine (Cr), indicating acute metabolic stress of CA-1 neurons during TGA [58]. The lactate peak was detected only in the DWI lesion and not in the perifocal tissue, suggesting that the metabolic changes in CA1 neurons were highly focal and not suggestive of a globally altered metabolic status in the hippocampus [58] (see Part I of this review for the pathogenetic implications of these neuroradiological findings).

#### 2.7.2. Positron Emission Tomography (PET) and Single-Photon Emission Computed Tomography (SPECT)

Many studies have investigated quantitative changes in regional cerebral glucose, oxygen metabolism, or cerebrovascular blood flow during TGA attacks by the use of PET or SPECT. However, these studies have provided controversial results regarding the type and site of the findings.

Most of the studies showed mesiotemporal alterations of perfusion or metabolism during the acute or post-acute phase of TGA [7,47,59,60,61,62,63,64,65,66,67,68,69,70].Many studies noted concomitant (decreased or increased) changes in cerebral blood flow in other anatomical structures, such as unilateral or bilateral thalamic, prefrontal, frontal, amygdalin, striatal, cerebellar, occipital, precentral, and postcentral areas [7,47,59,60,61,62,71,72,73,74,75,76,77,78].Several studies reported mono or bilateral hypoperfusion in the anatomical structures above mentioned [60,65,78,79,80,81,82].On the other hand, some studies showed increased perfusion in various limbic regions such as the hippocampus, amygdala, and thalamus [72,74,83,84].In some patients, hypoperfusion or hyperperfusion has been reported in the hippocampus, amygdala, and thalamus only of the left cerebral hemisphere [74,83,84,85].Finally, in some studies, no mesiotemporal or cerebral changes were detected [7,85,86].

In summary, the imaging data derived from PET and SPECT studies are difficult to compare and interpret. The variabilities in PET and SPECT findings are partly related to differences in the study designs, such as the imaging protocol and resolution. Most publications, in fact, are in the form of case reports or small cohorts in which the authors used visual or region of interest (ROI) inspection methods that limit the detection of subtle changes in regional cerebral blood flow (rCBF). Furthermore, different timing of scans after the TGA attack, frequently performed outside the time window needed to detect pathophysiological functional alterations, may further explain the inconsistency of findings among studies.

### 2.8. Differential Diagnosis

The differential diagnosis of TGA includes some disease states presenting with transient anterograde amnesia and sharing structural or functional alteration of memory circuits. Etiological diagnosis can be challenging and the underlying causes are diverse. However, an accurate collection of anamnestic and clinical data, as well as the use of suitable neuroradiological and instrumental investigations can support a differential diagnosis in the emergency department. The main features of some of these alternative diagnoses are described below and shortly reviewed on Table 3.

#### 2.8.1. Transient Epileptic Amnesia (TEA)

TEA is a form of adult-onset mesial temporal lobe epilepsy characterized by repeated episodes of transient amnesia [86,87,88,89]. The following features of TEA differ from TGA:Attacks occur more often during morning hours (70%), last less than 1 h and have a high recurrence rate [5,52,90].The amnesia during the events may be antero-retrograde or only retrograde [4,87].A large percentage of patients with TEA have an interictal mnemonic failure (predominantly of episodic memory), as well as patchy loss of autobiographical memory, and visuospatial memory decline [5,52,91].Interictal temporal lobe features such as oral automatisms, olfactory/gustatory hallucinations, epigastric aura, déjà vu, contact ruptures, etc., are frequent [5,92].Recording of EEG epileptiform activity is essential for diagnosis. However, between 30% and 43% of routine EEGs are normal [5,52,90].In the acute phase of the seizure, the majority of patients have normal MRI findings. However, in those with detectable brain signal abnormality on the MRI, there may be diffuse signal abnormality and swelling in the hippocampi [5,93]. Additional signal abnormalities can sometimes be seen in the pulvinar and/or cortex [93].The response to antiepileptic therapy is clear [94].

#### 2.8.2. Ischemic or Hypoxic Events

Ischemic events that occur in structures directly involved in mnemonic processes, such as the Papez’s circuit, or which, due to their specific location, cause a functional disconnection of large brain areas with consequent memory impairment, can mimic TGA [4,5]. In particular, unilateral, isolated infarction of the hippocampus or thalamus as well as ischemia within the medial temporal lobe (MTL) with involvement of the hippocampus, caudate, or fornix have been associated with a TGA-like presentation [4,95,96,97,98,99,100].

The etiology of hippocampal strategic stroke presenting with amnesia is generally cardioembolic (>50% cases) or related to large-artery disease of the vertebrobasilar system [5,101,102]. However, TGA-like symptoms have also been associated with events of a vasospastic nature or related to arterial insufficiencies such as vascular procedures requiring intravenous contrast use, (i.e., cerebral angiography), aortic dissection, and use of phosphodiesterase type 5 inhibitor [4,103,104,105,106,107,108,109,110].

The following considerations point to the vascular nature of the memory disorder:Patients with amnesia related to ischemic events usually do not present emotional or physical triggers prior to symptom onset, are older in age, and have known vascular risk factors [5,102,111].In ischemic events, focal neurological deficits are common, and amnesia may be associated with other minor cognitive symptoms such as executive deficit or mild anomia [4,5,102].At the onset of symptoms, brain MRI DWI and FLAIR imaging show acute changes in ischemic events while the same MRI sequences are usually negative in TGA [4].

In patients presenting with chest pain and cardiovascular changes such as hypotension and hypertension or asymmetric extremity blood pressures, CT angiography of the abdomen may be required to exclude aortic dissection.

#### 2.8.3. Migraine

Migraines and TGA share numerous characteristics, such as paroxysmal presentation, associated triggers and accompanying vegetative symptoms [18]. Furthermore, some studies have shown that TGA may occur in association with a migraine headache or exclusively as an isolated presenting aura [4,112,113,114,115]. In a French retrospective study, six cases of TGA occurring during a migraine attack were identified among 8821 new migraine patients [106]. TGA always occurred during a severe migraine attack, with vomiting efforts, so the authors hypothesized that a Valsalva maneuver, such as forceful vomiting, causing retrograde transmission of high venous pressure to the cerebral venous system, could be the major precipitating factor for TGA [106].

TGA presents some similarities, such as a transient–reversible nature and benign prognosis, with late-onset migraine accompaniments (LOMAs), a migraine syndrome characterized by transient neurological episodes mimicking TIAs [116]. Visual symptoms are the most common presentation, followed, respectively, by sensory, aphasic, and motor symptoms. LOMAs can occur without a headache and even without a prior history of migraine, especially in elderly individuals [19,116]. However, despite the obvious similarities between the two entities, some features, such as the stereotypic recurrences and the presence of focal neurological deficits, clearly support the diagnosis of LOMAs [19,116].

#### 2.8.4. Dissociative Amnesia (DA)

DA, also known as psychogenic amnesia, is characterized by retrograde amnesia of acute onset, limited to the episodic autobiographical domain, with variable loss of personal identity [5,117]. It is generally episodic and not usually accompanied by anterograde amnesia [5]. The average age at presentation is between 20 and 40 years and the distribution in men and women is similar [5,118]. DA is usually preceded by a traumatic trigger of a physical or psychological nature, and it is accompanied by a variable psychological reaction: some patients are very worried, while others express little concern over the symptom (a condition known as la belle indifference) [5,117]. Studies with 18-FDG PET in patients with DA show hypometabolism in right fronto-temporal areas [119].

#### 2.8.5. Toxic Amnesia

Some young patients with a history of illicit substance abuse, such as opioids and cocaine, have an acute onset of isolated anterograde amnesia. However, the majority of these patients have associated deficits in orientation, dysexecutive syndrome and varying degrees of sensory impairment [5]. Interestingly, a unique cluster of 14 cases of sudden onset amnesia with acute, complete, and bilateral ischemia of the hippocampus on brain MRI was identified in Massachusetts during 2012–2016 [120].

### 2.9. Management and Treatment

Below, we list some simple rules for TGA management and treatment.

Diagnosis and management of TGA should be performed by an interprofessional team, at least composed of a neurologist, internist, radiologist, and nurse practitioner, in order to properly address the large number of differential diagnoses that need to be considered when patients present with acute amnestic syndromes.The patient should be examined carefully for any accompanying neurological deficits or evidence of head trauma, both of which would exclude TGA as the diagnosis.Since ischemic stroke treatment is a time-dependent, sometimes invasive procedure, it is extremely important in an emergency setting to rule out ischemic stroke as a cause of amnesia to avoid subjecting the patient to unnecessary or even dangerous treatments. The early use of MRI and the detection of positive DWI lesions in the first hours after the onset of symptoms is essential to recognize the ischemic origin of the event. Further investigations, including extracranial and transcranial arterial echo-color Doppler sonography as well as ultrasound examination of the extracranial venous system, can be recommended if cerebrovascular risk factors are present and the patient is younger than 50 years [7].If the patient has repetitive amnestic episodes, EEG is mandatory to exclude a TEA.Transthoracic echocardiography may be indicated to evaluate septal hypertrophy, a marker of chronic hypertension, in TGA patients with elevated blood pressure on admission.The patient should be observed in the hospital until the memory deficit resolves.It seems prudent to avoid any activity that could raise intrathoracic venous pressure until the amnesia is resolved [3].Since TGA is a very stressful condition, it is necessary to guarantee psychological support not only for the patient but also for the family members, who also may require reassurance [32].Clinicians need to reassure the caregiver and patient about the benign nature of the disorder. TGA is a self-limiting condition that resolves spontaneously and rarely recurs.To date, there are no established, evidence-based treatments to prevent TGA recurrence. However, the recent demonstration of a strong association between acute hypertensive peaks and TGA in patients not adapted to chronic hypertension, i.e., without microangiopathy and septal hypertrophy, suggests that avoiding blood pressure peaks would be advisable [25]. In this regard, it is worth mentioning a case report of successful prophylaxis of recurrent coital TGA with accurate blood pressure control [121].Possible acute treatments or secondary prevention are indicated if an alternative diagnosis, (i.e., seizures or ischemic stroke/TIA) is made.Patients with risks of stroke or major cardiovascular events should be treated according to primary prevention guidelines.Patients should undergo periodic instrumental checks, (e.g., EEG, cardiological examinations, etc.), but do not need any restrictions in daily activities once the memory deficits have been resolved.

## 3. Long-Term Outcome

TGA is considered a benign condition that, according to the standard accepted criteria, resolves within 24 h [1,2,3,4]. However, in recent years, many studies have analyzed the long-term outcome of patients with TGA regarding the risk of recurrence and of later developing complications such as cognitive impairment, stroke, and seizures. Below are the main results of these studies.

### 3.1. Recurrence Rate for TGA

The annual rate of TGA recurrence reported in different studies varies from 2.9 to 26.3%, depending on the length of follow-up and the sensitivity of the definition used to identify cases of TGA [122,123,124,125,126,127]. A study on 93 prospective patients with TGA found that risk factors associated with recurrence were head injury, depression, and family history of dementia [127]. In a subsequent systematic review of 1989 patients with TGA, 13.5% of cases experienced a recurrence of TGA and results were suggestive of a relationship between recurrence and a family or personal history of migraines, as well as a personal history of depression. However, there was only a weak relationship between recurrence and a family history of dementia and a personal history of head injury [128]. In a recent study of 340 German TGA patients with a follow-up of at least five years, patients with TGA recurrence were significantly younger and had less extent of cerebral microangiopathy than patients with isolated TGA episodes [129]. Specifically, patients up to 70 years of age without microangiopathic changes on MRI (Fazekas’ score 0) had a 24.5% risk of subsequent TGA recurrence over the next five years [129].

Finally, it is worth noting that the duration of symptoms does not appear to influence the long-term prognosis of TGA. Indeed, in a study of 639 patients with TGA, there were no differences in the long-term risk of seizures/epilepsy or major cardiovascular events between TGA lasting <1 h and TGA lasting ≥1 h [36].

### 3.2. Cognitive Profile of TGA Patients after the Acute Phase

The data in the literature on this topic are quite discordant. Many studies suggested complete recovery of cognitive function after a TGA episode [3,6,37,38,39,40,48]. In agreement with these studies, a meta-analysis comparing 374 patients with TGA to 760 control subjects found no significant differences between patients and controls in any cognitive domain (anterograde episode long-term memory, retrograde episode long-term memory, short-term memory, semantic memory, working memory, attention and executive function) under investigation [48]. Furthermore, in a study on seventeen well-defined TGA cases, psychometric evaluation with a comprehensive neuropsychological test battery, carried out 2 years after the attack, revealed no differences in cognitive performance between patients and HC [130]. On the other hand, the results of other studies indicated that subtle memory reductions may persist in the long-term phase, suggesting that there is only an incomplete recovery of memory function [42,43,45,48,131,132].

### 3.3. Long-Term Risk of Developing a Cognitive Decline

To date, only a few data are available about the long-term risk of developing a cognitive decline. Borroni et al. submitted fifty-five patients to a standardized neuropsychological assessment one-year after the TGA attack and found that 18 out 55 (32.7%) fulfilled the criteria for amnestic mild cognitive impairment [45]. Furthermore, in a study on 102 TGA patients with an average follow-up of 82.2 months, only 3 cases of dementia were recorded. Thus, at the end of follow-up, the prevalence was about 2.9% which is similar to that of the general population [123].

### 3.4. Risk of Ischemic Stroke

Clinical studies looking at the risk of stroke after TGA have reported conflicting results. Some previous studies of retrospective design and not using standardized diagnostic criteria for TGA suggested that up to 46% of TGA patients had a stroke during a follow-up of 2 to 7 years [133,134]. However, most of the subsequent studies have shown that patients with TGA have a similar risk of stroke, myocardial infarction, and peripheral artery disease as the general population [2,13,16,123,135,136]. A study by Mangla and coworkers including 4299 patients with TGA demonstrated that the stroke risk after the diagnosis of TGA (0.54%) was similar to that after the diagnosis of migraine (0.22%) and was much lower than that after the diagnosis of TIA (4.72%) [111]. In agreement with these results, a study on 525 TGA patients found a pooled annual stroke risk of 0.6% (95% CI, 0.4–0.9), demonstrating that TGA does not carry an increased risk of stroke or major cardiovascular events, at least when cardiovascular risk factors are treated according to primary prevention guidelines [137]. Finally, further confirming these data, a recent nationwide population cohort study, which analyzed 27,266 hospitalizations with a primary discharge diagnosis of TGA, found that the risk of readmission for ischemic stroke in patients with TGA was no different compared with the control group (HR: 1.13, 95% CI 0.62–2.05, P 0.686) during the mean (SD) follow-up period of 192.2 (102.4) days [138].

### 3.5. Risk of Epilepsy

No less discordant are the results of the studies on the risk of epilepsy after TGA. Hodges et al. in two studies on 153 and 114 TGA cases, respectively, found that 7% of patients developed epilepsy on follow-up [1,2]. A population-based study of 221 consecutive TGA cases with a mean 12-year follow-up, conducted by Arena et al. in 2017, could not find a significant difference between TGA and non-TGA controls in seizures [16]. However, a recent control cohort study with an 8-year follow-up period on 185 TGA subjects and 555 non-TGA controls, showed that TGA is associated with increased long-term risk of epilepsy. In this study, there were seven subjects developing epilepsy in the TGA cohort during the follow-up period with a yearly incidence rate of 9.629 per 1000 persons. The adjusted hazard ratio for developing epilepsy in the TGA cohort was 6.50 (95% CI, 1.87–22.68, *p* = 0.003) compared with the non-TGA cohort [139].

## 4. Diagnostic Flow Chart

As patients commonly are referred to the emergency room doctors or family physicians, who may sometimes be unfamiliar with this unusual diagnosis, it is very important to have a quick and clear clinical path to avoid inappropriate investigation and management of these patients.

Diagnostic criteria used so far no longer appear to be a reliable diagnostic tool as they do not contemplate the possibility that patients with TGA may have some degree of retrograde amnesia or the involvement of other cognitive functions during the episode. Furthermore, a thorough evaluation of TGA cannot be separated from some instrumental investigations, such as MRI performed 24–96 h after the onset of symptoms. Therefore, we propose a pragmatic flow chart to support clinicians in the diagnostic process (Figure 3).

## 5. Conclusions

TGA is a temporary memory loss syndrome that still remains enigmatic due to its etiology and clinical features. The diagnosis is mainly clinical and requires the exclusion of some acute amnestic syndromes that share a temporary deficit of the memory system, secondary to generalized brain dysfunction or focal hippocampal damage. A standardized diagnostic approach may help clinicians manage this disease and avoid misdiagnosis. Finally, although the clinical course of TGA is self-limited and only occasionally mild memory impairment may persist for weeks, little is known about the long-term prognosis. Available data suggest that TGA does not affect the future risk of cerebrovascular events, but more information is needed regarding the long-term risk of epilepsy and, particularly, cognitive decline.

## Figures and Tables

**Figure 1 jcm-11-03940-f001:**
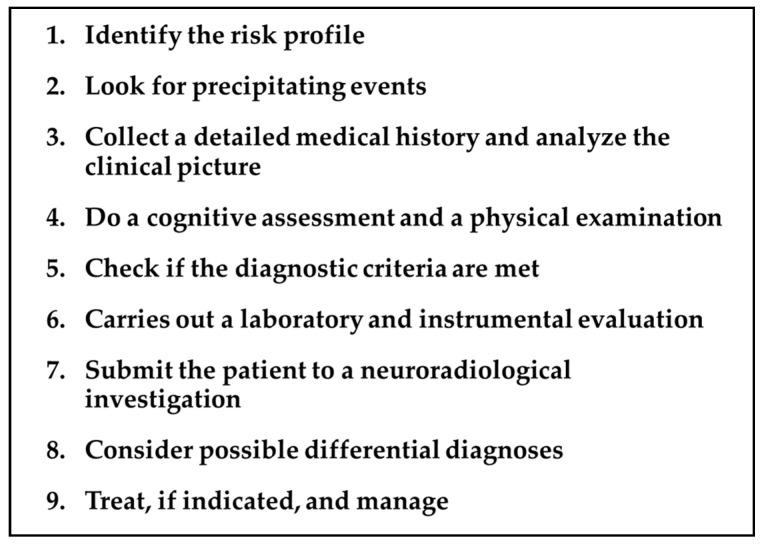
TGA roadmap.

**Figure 2 jcm-11-03940-f002:**
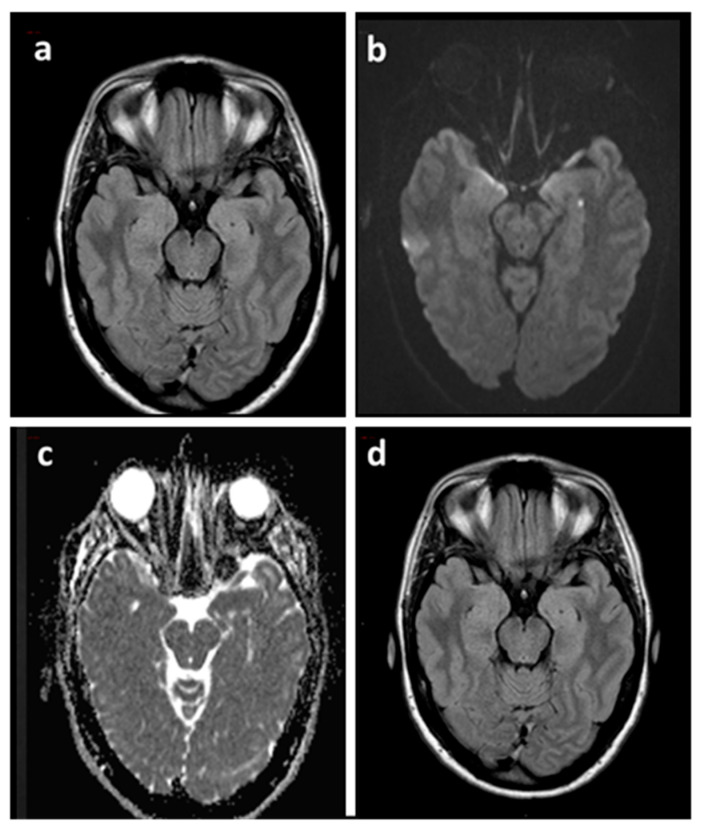
Brain MRI of a TGA patient acquired 48 h after the episode (**a**–**c**) and at three months (**d**). Axial FLAIR sequence (**a**) showing both hippocampi without signal abnormalities corresponding to the punctate DWI hyperintensity (**b**) with ADC hypointensity (**c**) in the left hippocampus. Axial FLAIR sequence at three months (**d**) confirmed the absence of signal abnormalities. ADC: Apparent Diffusion Coefficient.

**Figure 3 jcm-11-03940-f003:**
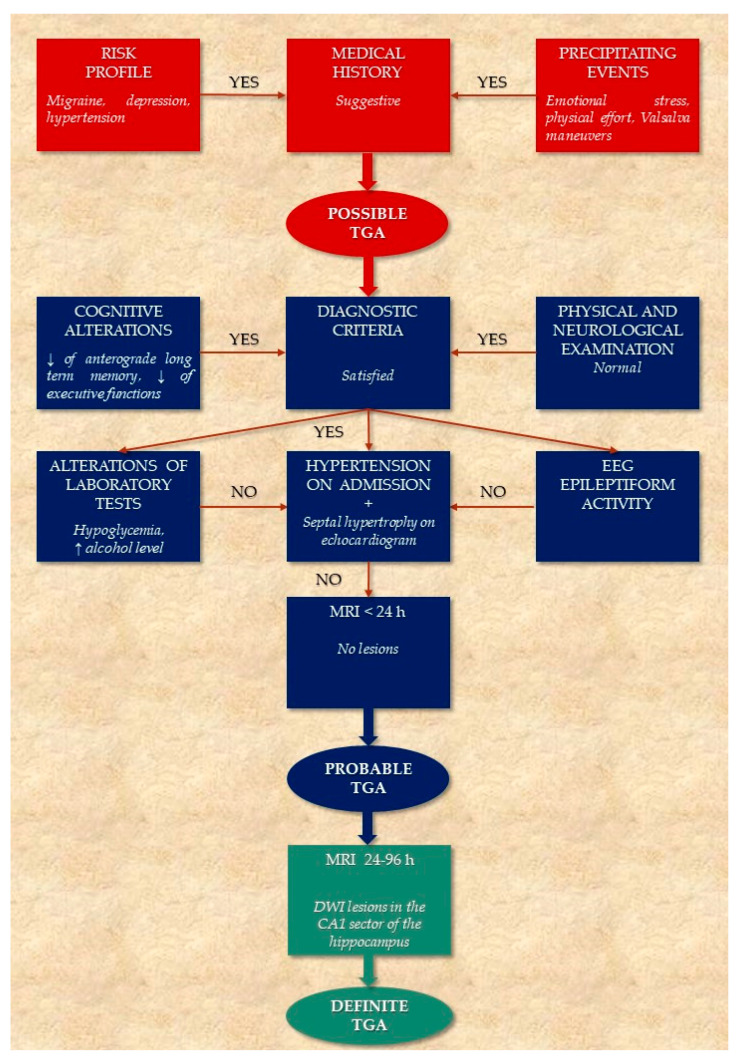
Algorithm for the diagnostic workup of TGA.

**Table 1 jcm-11-03940-t001:** Diagnostic criteria for transient global amnesia (data from Hodges JR and Warlow CP 1990) [1,2].

Main Diagnostic Features of TGA
Attack must be witnessed
There must be anterograde amnesia during the attack
Cognitive impairment limited to amnesia
No clouding of consciousness or loss of personal identity
No focal neurological signs/symptoms
No epileptic features
Attack must resolve within 24 h
No recent head injury or active epilepsy

**Table 2 jcm-11-03940-t002:** MRI findings and parameters in TGA.

Main MRI Issues in TGA
DWI-MR are the optimal sequences for detecting brain lesions in TGA;
Lesions are selectively found in the CA1 sector of the hippocampal cornu ammonis of one or both sides;
Lesions can be single or multiple and vary in size from 1 to 5 mm;
DWI sequences with a slice thickness ≤ 3 mm have the higher diagnostic yield;
Both 3 T and 1.5 T imaging have a similar diagnostic yield;
The best time window for the detection of hippocampal lesions is between 24 and 96 h after symptom onset;
Hippocampal DWI lesions generally resolve 7–10 days after onset of TGA, with no long-term persistent signal changes on T2 or FLAIR sequences.

DWI: Diffusion Weighted Imaging; MR: Magnetic Resonance; T: Tesla; TGA: Transient Global Amnesia; FLAIR: Fluid Attenuated Inversion Recovery.

**Table 3 jcm-11-03940-t003:** Differential diagnosis of TGA (modified from Arena JE, et al., 2015 and Spiegel DR et al., 2017) [1,5].

Condition	Risk Factors	Precipitating Factors	Duration	Associated Neurologic Symptoms	MRI	EEG	Recurrence	Response to Anti-Epileptics
TGA	Migraine	Yes	4–6 h	No	Transient hippocampal DWI hyperintensity	Normal	Low	No
TEA	No	No/yes	<60 min	No/yes (automatisms, olfactory or gustatory hallucinations)	Normal/hippocampal sclerosis or atrophy	Abnormal	High	Yes
Ischemic events	Vascular	No	Minutes to permanent impairment	No/yes (any)	DWI with T2-FLAIR permanent lesion	Normal	Low	No
Hypoxic events (i.e., aortic dissection)	Increased intrathoracic pressure	Stress reaction due to pain	10–12 h	No	Normal	Normal	Not known	No
Migraine	Genetic, dietary	Yes (fasting, emotional stress, sleep problems)	4–72 h	Auras up to 30% (visual, sensory, motor, or language abnormalities)	Normal	Normal	High	Yes
DA	Trauma	Yes emotional stress	Variable	No	Normal	Normal	Varies	No
Toxic amnesia	Substance abuse	No	Variable	Yes (disorientation, dysexecutive syndrome, etc.)	Normal or bilateral hippocampal ischemia	Normal	High	No

Abbreviations: TGA = transient global amnesia; TEA = transient epileptic amnesia; DA = dissociative amnesia; MRI = magnetic resonance imaging; DWI = diffusion-weighted; FLAIR = fluid-attenuated inversion recovery; EEG = electroencephalography.

## Data Availability

Not applicable.

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
