# Peer review of "Forgetting the Unforgettable: Transient Global Amnesia Part II: A Clinical Road Map"

_jcm, 2022, doi:10.3390/jcm11143940_

Round 1

Reviewer 1 Report

The manuscript presented is a useful clinical practice roadmap for the care of acute patients with transient global amnesia (TGA). Despite many cases described, the diagnosis of this condition has pitfalls, especially in the emergency setting. The manuscript is helpful in structuring the clinical pathway to confident diagnosis.

Regarding the vascular risk profile (page 3, lines 87 ff), I think a more detailed presentation of the relationship of acute blood pressure peaks in patients without evidence of chronic hypertension would be helpful, as the relationship has been detailed in a recent study. Patients with TGA have higher blood pressures on admission compared with stroke patients but lower signs of chronic hypertension as measured by the extent of cerebral microangiopathy and cardiac hypertrophy (Rogalewski et al.; doi: 10.3389/fneur.2021.666632).

With regard to precipitating events, it would be advisable to also list interpersonal conflicts under the heading of emotional stress, especially since these were identified as frequent in a recent study (Hoyer et al.; doi: 10.1016/j.jns.2021.117464). In addition, it seems desirable to go into a little more detail about possible differences between females and males (Hoyer et al.; doi: 10.1016/j.jns.2021.117464).

In the field of additional examinations (page 5, lines 182 ff), transthoracic echocardiography could be discussed in case of an initial acute hypertensive crisis to differentiate chronic hypertension from acute hypertension. At the very least, this may already detect septal hypertrophy in some patients with previously unknown hypertension and thus support the indication for antihypertensive medication.

In Table 2, I disagree with the point "The addition of T2-weighted images to DWI is of little or no utility in the follow-up of patients with TGA." (page 6, lines 233). First, assessment of the extent of cerebral microangiopathy by additional sequences (e.g. FLAIR sequences) provides a measure of the presence of chronic hypertension. This allows the conclusion of a stricter treatment regime by means of antihypertensive medication. In addition, the absence of cerebral microangiopathy has been described as a potential predisposing factor for TGA recurrences. A 24.5% risk of subsequent TGA recurrence in the following five years was determined for TGA patients up to 70 years of age without microangiopathic changes on MRI (Fazekas’ score 0) (Rogalewski et al.; doi: 10.3389/fneur.2021.736563). Especially the assessment of a risk of recurrence is a frequent issue in the treatment of patients with TGA and can be estimated using such data.

I find Table 3 very helpful in distinguishing the various differential diagnoses.

Regarding TGA recurrences, the aforementioned association with age as well as the extent of cerebral microangiopathy could be mentioned. Overall, no evidence for treatment can be derived in the prevention of TGA recurrence. Interestingly, case reports suggesting stricter blood pressure control exist and could be mentioned as anecdotal reports (Berlit P.; doi: 10.1212/WNL.55.12.1937).

Regarding the diagnostic flow chart, a complementary mention would be useful that patients often have hypertensive blood pressures (Rogalewski et al.; doi: 10.3389/fneur.2021.666632). This does not contradict the diagnosis of probable TGA, but possibly supports it. However, it seems reasonable to pay attention to this and to treat these aberrations as well.

Author Response

We would like to thank the reviewer for his/her comments and suggestions which allow us to improve the manuscript. We have modified the paper accordingly and there are our answers to the suggestions.

The manuscript presented is a useful clinical practice roadmap for the care of acute patients with transient global amnesia (TGA). Despite many cases described, the diagnosis of this condition has pitfalls, especially in the emergency setting. The manuscript is helpful in structuring the clinical pathway to confident diagnosis.

Many thanls for appreciating our paper.

Regarding the vascular risk profile (page 3, lines 87 ff), I think a more detailed presentation of the relationship of acute blood pressure peaks in patients without evidence of chronic hypertension would be helpful, as the relationship has been detailed in a recent study. Patients with TGA have higher blood pressures on admission compared with stroke patients but lower signs of chronic hypertension as measured by the extent of cerebral microangiopathy and cardiac hypertrophy (Rogalewski et al.; doi: 10.3389/fneur.2021.666632).

We incorporated the suggested paper among the references and assed a couple of sentences detailing more deeply the role of arterial hypertension depending of preexisting history of disease.  

With regard to precipitating events, it would be advisable to also list interpersonal conflicts under the heading of emotional stress, especially since these were identified as frequent in a recent study (Hoyer et al.; doi: 10.1016/j.jns.2021.117464). In addition, it seems desirable to go into a little more detail about possible differences between females and males (Hoyer et al.; doi: 10.1016/j.jns.2021.117464).

Thanks for the suggestion. We modified the text and the references accordingly. 

In the field of additional examinations (page 5, lines 182 ff), transthoracic echocardiography could be discussed in case of an initial acute hypertensive crisis to differentiate chronic hypertension from acute hypertension. At the very least, this may already detect septal hypertrophy in some patients with previously unknown hypertension and thus support the indication for antihypertensive medication.

We added a short paragraph detailing more deeply the role of TTE in the work-up of TGA.

In Table 2, I disagree with the point "The addition of T2-weighted images to DWI is of little or no utility in the follow-up of patients with TGA." (page 6, lines 233). First, assessment of the extent of cerebral microangiopathy by additional sequences (e.g. FLAIR sequences) provides a measure of the presence of chronic hypertension. This allows the conclusion of a stricter treatment regime by means of antihypertensive medication. In addition, the absence of cerebral microangiopathy has been described as a potential predisposing factor for TGA recurrences. A 24.5% risk of subsequent TGA recurrence in the following five years was determined for TGA patients up to 70 years of age without microangiopathic changes on MRI (Fazekas’ score 0) (Rogalewski et al.; doi: 10.3389/fneur.2021.736563). Especially the assessment of a risk of recurrence is a frequent issue in the treatment of patients with TGA and can be estimated using such data.

We agree with the comment of the reviewer and we changed the text adding more details about the role of MRI in the asessment of SVD signs.

I find Table 3 very helpful in distinguishing the various differential diagnoses.

Many thanks again.

Regarding TGA recurrences, the aforementioned association with age as well as the extent of cerebral microangiopathy could be mentioned. Overall, no evidence for treatment can be derived in the prevention of TGA recurrence. Interestingly, case reports suggesting stricter blood pressure control exist and could be mentioned as anecdotal reports (Berlit P.; doi: 10.1212/WNL.55.12.1937).

Thanks for the suggestion. We added this reference and the message suggested by this case series in the text.

Regarding the diagnostic flow chart, a complementary mention would be useful that patients often have hypertensive blood pressures (Rogalewski et al.; doi: 10.3389/fneur.2021.666632). This does not contradict the diagnosis of probable TGA, but possibly supports it. However, it seems reasonable to pay attention to this and to treat these aberrations as well.

We modified the flowchart adding the suggested item.

Reviewer 2 Report

In this review, the authors detail the clinical approach and long-term outcomes of transient global amnesia (TGA). The article is well-written and summarizes the current knowledge regarding this topic very well. I have a few minor suggestions:

1. Consider adding recent data regarding the association between the duration of symptoms and the long-term prognosis of TGA.

2. In section 3.4 [Risk of Ischemic Stroke], consider citing the recent study including a large number of TGA patients that showed that TGA is not associated with an increased risk of ischemic stroke as compared to the general population [PMID 33651152].

Author Response

In this review, the authors detail the clinical approach and long-term outcomes of transient global amnesia (TGA). The article is well-written and summarizes the current knowledge regarding this topic very well.

Many thanks for the appreciation of our paper.

I have a few minor suggestions:

  1. Consider adding recent data regarding the association between the duration of symptoms and the long-term prognosis of TGA.

Thanks for the suggestion. We added a sentence in the corresponding paragraph about this issue.

  1. In section 3.4 [Risk of Ischemic Stroke], consider citing the recent study including a large number of TGA patients that showed that TGA is not associated with an increased risk of ischemic stroke as compared to the general population [PMID 33651152]

Thanks again for the suggestion. We added the suggested paper and a corresponding sentence in the text.